# Environmental DNA Metabarcoding Reveals Divergent Patterns of Biodiversity, Community Assembly, and Environmental Sensitivity Across Taxa in Adjacent Rivers

**DOI:** 10.3390/biology14121796

**Published:** 2025-12-17

**Authors:** Yimei Wei, Jingwei Zhang, Shuping Wang, Zhenjun Tian, Xiaolong Lin, Yongkun Yu, Yangwei Bai

**Affiliations:** 1State Key Laboratory of Environmental Criterion and Risk Assessment, Chinese Research Academy of Environmental Sciences, Beijing 100012, China; wei.yimei@craes.org.cn (Y.W.); m240100101@st.shou.edu.cn (J.Z.); wangsp@craes.org.cn (S.W.); tianzj@craes.org.cn (Z.T.); yyk1169432061@163.com (Y.Y.); 2Key Laboratory of Estuarine and Coastal Environment, Ministry of Ecology and Environment, Beijing 100012, China; 3Engineering Research Center of Environmental DNA and Ecological Water Health Assessment, Shanghai Ocean University, Shanghai 201306, China; xllin@shou.edu.cn

**Keywords:** eDNA, river ecosystems, community assembly, assembly mechanisms, neutral community model

## Abstract

Understanding how different biological groups respond to environmental changes is crucial for river ecosystem management. This study examined two neighboring rivers with different environmental conditions using environmental DNA to analyze the plankton and fish communities. While the overall species richness was similar between the rivers, the community composition varied significantly, indicating high spatial turnover. Both groups exhibited a distance-decay relationship, with plankton showing a more random community assembly, whereas fish communities were more influenced by specific environmental factors. The variation in both communities was linked to physical and chemical factors such as conductivity, pH, temperature, oxygen levels, and pollution, with plankton also responding to nutrient levels like phosphorus and nitrogen. These findings highlight the distinct environmental selection processes shaping different biological groups, offering insights for targeted river restoration and water quality improvement.

## 1. Introduction

River ecosystems comprise a complex combination of hydrological and geomorphological features, nutrient dynamics, and biological communities [1]. With continued global population growth, the expansion of certain land uses, such as agricultural intensification and urbanization, has exacerbated a range of environmental problems, including water pollution, overexploitation of water resources, and destruction of riparian vegetation. These pressures collectively accelerate the degradation of river ecosystems [2,3]. Regarding urbanization, land use patterns directly or indirectly reflect the profound influences of human activities on river water quality, aquatic ecological functions, and biodiversity [4]. Previous studies have shown that land use types subject to intense human disturbance, such as farmland and urban construction land, significantly impact water pollution loads [5]. Multiple studies have further revealed significant or highly significant negative correlations between aquatic biodiversity across multiple biological groups and anthropogenic land use (e.g., cropland and impervious surfaces). Environmental variables such as dissolved oxygen, total phosphorus, and pH significantly regulate the spatial distribution and composition of aquatic communities [6]. These findings illustrate the mechanism through which land use indirectly shapes aquatic community structure by altering environmental conditions.

Aquatic organisms differ greatly in ecological niches, dispersal capacity, body size, and sensitivity to disturbance, resulting in heterogeneous responses to environmental stressors [7]. These differences suggest that community assembly mechanisms may vary substantially among biological groups. Deterministic processes, such as environmental filtering and biotic interactions, are emphasized through niche theory [8], whereas neutral theory focuses on stochasticity, including dispersal limitation and ecological drift [9]. Contemporary studies recognize that both forces operate simultaneously, with their relative importance depending on organismal traits and habitat conditions [10,11]. However, despite strong theoretical expectations, empirical studies comparing how different biological groups assemble and respond to environmental gradients in river ecosystems remain limited.

In this study, we categorized key river biota into plankton (phytoplankton and zooplankton) and fish. As primary producers in river ecosystems, phytoplankton play a pivotal role in energy flow and material cycling [12]. Zooplankton (including cladocerans, copepods, protozoa, and rotifers) feed on phytoplankton, bacteria, and other microorganisms. Due to their minute size and high metabolic rates, eukaryotic plankton significantly contribute to the decomposition of organic matter and material cycling in aquatic environments [13]. Although phytoplankton and zooplankton occupy different trophic levels, they share common characteristics, such as small body size, short generation times, and high sensitivity to environmental changes. Consequently, they are often analyzed collectively as a “plankton functional group” in many ecological assessment and monitoring studies [14]. As top consumers of fish, fish not only play a crucial role in maintaining the food web structure and ecological balance but also serve as important biological indicators for assessing water quality and ecosystem health [15]. Current research on plankton primarily focuses on the ecological assessment and dynamic monitoring of diverse aquatic environments, including lakes, reservoirs, and rivers. Within an ecological functionalism framework, scientists analyze plankton data to determine key traits such as ecological characteristics, environmental adaptability, and functional roles, thereby enabling functional group classification. This methodology facilitates a comprehensive analysis of ecological functions and responses to environmental changes [16,17]. Fish research has predominantly addressed taxonomic diversity, functional traits, and responses to environmental gradients [18]. Although plankton and fish are fundamental components of freshwater ecosystems, there is a lack of research on their contrasting responses to anthropogenic pressures and environmental changes across trophic levels. Conducting comparative studies is paramount, as they help to elucidate potential disparities in community assembly mechanisms and environmental responses, thereby underscoring the need for comprehensive ecological investigations that span multiple biological groups within river systems.

In recent years, environmental DNA (eDNA) metabarcoding has advanced rapidly as an innovative tool for biodiversity monitoring. This technique is characterized by its non-invasiveness, high sensitivity, and capacity to detect multiple taxonomic groups simultaneously, making it particularly suitable for capturing biological information across different trophic levels. In aquatic ecosystems, eDNA has been extensively applied in species surveys, biomass estimation, and community structure analysis, demonstrating significant potential for ecological monitoring and environmental assessment [19,20]. The Tuhai River and the Chao River, located within China’s Yellow River Delta High-Efficiency Ecological Economic Zone, were selected for this study. Despite their geographical proximity and similar ecological functions as estuarine rivers, they exhibit markedly distinct land use patterns. The Tuhai River serves as a vital source for urban flood control, drainage, and ecological water supply, whereas the Chao River functions as an artificially excavated drainage channel flowing directly into the Bohai Sea. The two river systems differ significantly in terms of the intensity of anthropogenic disturbances, such as the proportion of land use made up by farmland and impervious surfaces, as well as in terms of water quality pressures. These differences provide an ideal setting for investigating the impacts of land use and water environment changes on aquatic biotic communities. This study utilizes eDNA metabarcoding, considering significant disparities in land use and water quality. Its objectives are as follows: (1) to assess the similarities and differences in aquatic biodiversity and community assembly processes across different biological groups in two adjacent rivers and (2) to determine how the environmental response sensitivity of aquatic communities differs across these biological groups.

## 2. Materials and Methods

### 2.1. Sample Collection and Acquisition of Spatial and Environmental Variables

This study focuses on the Chao River (117.8900° E–118.2400° E, 37.3417° N–38.0617° N) and the Tuhai River (117.5867° E–118.1233° E, 37.3067° N–38.0900° N), located in China’s Yellow River Delta. A total of 30 sampling sites were established, with 15 along the Chao River and 15 along the Tuhai River. Sampling was conducted in April 2024 (Figure 1).Three replicate samples and one blank control were collected at each sampling point. During sampling, one liter of water was collected from approximately 0.5 m below the ground surface for eDNA analysis, and an additional liter was obtained for assessing water chemistry parameters.

Land use data for riparian zones were obtained using ArcGIS 10.2. The proportions of each land use type within a 2 km radius around each sample point were extracted using the buffer analysis tool. The 2 km buffer zone radius employed in this study effectively captures the ecological connectivity within the research area. Land cover categories included cropland, impervious surfaces, water bodies, and others. In situ water quality parameters were measured using a Manta+ multiparameter water quality meter (Eureka, CA, USA); these parameters included electrical conductivity (EC), water temperature (WT), dissolved oxygen (DO), and the potential of hydrogen (pH) (GB3838-2002) [21]. The remaining water quality parameters were measured using standard laboratory analytical methods (Appendix A).

### 2.2. eDNA Extraction, PCR Amplification, and High-Throughput Sequencing

All water samples were filtered immediately after collection through 0.22 μm pore size, 142 mm diameter polycarbonate membranes (Millipore, Bedford, MA, USA) for concentration. The filtered membranes were transferred to sterile cryogenic vials and stored on site at −80 °C. All eDNA samples and blank controls were analyzed parallelly at each step of the experiment. Subsequently, they were transported to the laboratory and stored at −20 °C until DNA extraction. eDNA extraction was performed using the DNeasy PowerWater Kit (QIAGEN, Hilden, Germany). The extraction procedure adhered to the manufacturer’s protocol, including instructions for performing lysis, magnetic bead binding, washing, and elution. DNA concentration was quantified using Qubit^®^ 4.0, and integrity was verified via 2% agarose gel electrophoresis. The extracted DNA served as the template for PCR amplification with the following universal primers: MiFish-F (5′-GTCGGTAAAACTCGTGCCAGC-3′) and MiFish-R (5′-CATAGTGGGGTAT CTAATCCCAGTTTG-3′) for the 12S rRNA gene [22] and 18S V9F (5′-CCCTGCCHTTTGTACACAC-3′), and 18S V9R (5′-CCTTCY GCAGGTTCACCTAC-3′) for the 18S rRNA gene [23]. First-round PCR was carried out in a 30 μL reaction system containing a 30 μL reaction system consisting of 15 μL of 2× Hieff^®^ Robust PCR Master Mix, 1 μL of each primer (10 μM), 1 μL DNA template (10 ng), and 12 μL H_2_O. The thermal cycling protocol consisted of initial denaturation at 94 °C for 3 min, followed by 5 cycles of denaturation at 94 °C for 30 s, annealing at 45 °C for 20 s, and extension at 65 °C for 30 s; this process was then followed by another 5 cycles of denaturation at 94 °C for 20 s, annealing at 55 °C for 20 s, and extension at 72 °C for 30 s, with a final extension at 72 °C for 5 min. Each sample was amplified in three technical replicates, which were pooled to reduce PCR bias. The pooled PCR products were then visualized on 2% agarose gel. amplicons served as templates for the second PCR, where Illumina bridge PCR-compatible primers were introduced. The second-round PCR was performed in a 30 μL mixture containing 15 μL of 2× Hieff^®^ Robust PCR Master Mix, 1 μL of each primer (10 μM), 1 μL of purified product (20–30 ng), and 12 μL of H_2_O. The cycling conditions included initial denaturation at 95 °C for 3 min, 8 cycles of denaturation at 94 °C for 20 s, annealing at 55 °C for 20 s, and extension at 72 °C for 30 s, ending with a final extension at 72 °C for 5 min. Finally, high-throughput sequencing was conducted using the Illumina NovaSeq 6000 PE250 platform (Illumina, San Diego, CA, USA).

### 2.3. Bioinformatic Analysis

FASTQ format raw sequencing reads generated using Illumina HiSeq (Illumina, San Diego, CA, USA) ere initially processed to remove platform-specific adapters and PCR primers using Cutadapt, allowing up to 2 mismatches and requiring a minimum overlap of 10 bp. Reads were assigned to individual samples based on sample-specific barcode sequences, with no mismatches allowed, and sequence orientations were corrected. Paired-end reads were subsequently merged using PEAR, with a minimum overlap of 15 bp and a maximum mismatch ratio of 0.1. Low-quality bases at the 3′ ends were trimmed using PRINSEQ (v0.20.4) with a 10 bp sliding window, with reads trimmed when the mean Phred score within the window was below Q20. Reads that contained ambiguous bases or low-complexity sequences (entropy < 0.5) or were shorter than 150 bp (12S) or 200 bp (18S) were discarded to generate high-quality clean reads. Operational Taxonomic Units (OTUs) were constructed following the UPARSE workflow implemented in USEARCH. Quality-filtered reads were first dereplicated within and across samples, and singletons were removed. Chimeric sequences were identified and removed using UCHIME in reference mode against the SILVA 138 database (18S) for plankton and the MitoFish v3.73 database (12S) for fish. The remaining non-chimeric sequences were clustered into OTUs at 97% sequence similarity, with the most abundant sequence in each OTU selected as the representative samples. All filtered reads were then mapped back to OTU representatives at 97% identity to generate OTU abundance table. The taxonomic assignment of representative OTU sequences was performed using BLASTn (https://blast.ncbi.nlm.nih.gov/Blast.cgi (accessed on 15 April 2025)), with a minimum identity of 97% and query coverage of 90%, against the SILVA 138 database for plankton 18S sequences and the MitoFish v3.73 database for fish 12S sequences, aiming to reduce misclassification. The relative abundances of OTUs were calculated as the percentages of total reads in each sample.

### 2.4. Statistical Analysis

#### 2.4.1. α- and β-Diversity Analysis

To evaluate species diversity levels across trophic levels in each sample, five widely used α-diversity indices were computed: Chao1, Richness, the Shannon index, the Simpson index, and Pielou’s evenness. The Chao1 index estimates the number of potentially unobserved OTUs in a community, reflecting its potential species richness. Richness represents the actual number of observed OTUs in a sample. The Shannon index integrates both species richness and evenness, serving as a common metric for overall community diversity. The Simpson index emphasizes the influence of dominant species, embodying the principle of quantifying dominance to assess diversity. Higher values indicate stronger dominance and lower diversity. The Pielou’s evenness quantifies the uniformity of species abundance distribution, with values closer to 1 reflecting a more balanced community structure. This aligns with classical ecological objectives—distinguishing species richness from abundance distribution uniformity—thereby enabling a more comprehensive understanding of community organization patterns. All diversity indices were calculated via R (v4.3.2) using the vegan package.

#### 2.4.2. Analysis of Community Assembly Mechanisms and Environmental Responses

To explore the ecological mechanisms governing community assembly, the Neutral Community Model (NCM) developed by Sloan et al. [24] was applied. Key model parameters include R^2^ (reflecting model fit) and Nm (the product of the migration rate (N) and the migration probability (m)), which help to quantify the relative importance of stochastic processes (e.g., ecological drift, diffusion constraints) versus deterministic processes (e.g., environmental filtering, interspecific interactions) in shaping community structure. The neutral model was implemented following the analytical framework established by Lu et al. [1] within the R environment. Community β-diversity patterns were visualized through principal coordinate analysis (PCoA) based on the Bray–Curtis dissimilarity. A distance decay curve was generated to examine the relationship between geographic distance and community similarity, thereby illustrating the spatial turnover rate of species. All computations and visualizations were performed in R using packages including vegan (v.2.6–2), ape (v5.7-1), and ggplot2 (v3.5.1). To evaluate the influence of aquatic environmental factors on community structure, canonical correspondence analysis (CCA) was separately performed on fish and plankton communities, revealing how community distribution responds to gradients in water quality. Prior to performing CCA, all environmental variables were standardized using SPSS software (v25.0). Additionally, a Mantel test was performed to assess the correlation between the community dissimilarity matrix and nine water quality parameters (EC, WT, DO, pH, TP, TN, NH_4_-N, COD, BOD_5_). The Mantel test was implemented using the ‘mantel’ function from the vegan package, employing Spearman’s correlation with 999 permutations.

## 3. Results

### 3.1. Differences in Land Use and Water Environment Between Adjacent Rivers

A two-kilometer buffer zone was delineated along each riverbank to quantify the land use composition. Marked differences in land cover types were observed between the Tuhai River and the Chao River (Appendix A). Farmland was the dominant land use type along both rivers, making up 67.47% of the total buffer area for the Chao River and 74.87% for the Tuhai River. The proportion of impervious surfaces was slightly higher along the Chao River (20.78%) than along the Tuhai River (18.65%). Both rivers displayed minimal coverage of grassland and bare land, with each constituting less than 1% of the buffer area. Overall, the Tuhai River exhibited significantly greater proportions of farmland and impervious surfaces than the Chao River, reflecting more intensive agricultural and urban land use. Natural landform types made up very low proportions in both river systems. PCoA based on the Bray–Curtis distance revealed a clear difference in water quality parameters between the two rivers (Figure 2). The Tuhai River exhibited higher concentrations of EC, COD, and BOD_5_, whereas the Chao River generally exhibited elevated levels of TN, NH_4_-N, and TP (Appendix A).

### 3.2. Community Composition and α-Diversity Based on eDNA Metabarcoding

A total of 322,488 sequences and 1,535,181 sequences were ultimately retained for fish and plankton eDNA metabarcoding, respectively. The accurate OTUs for both plankton and fish were retained. Non-target sequences such as environmental background contaminants were excluded during OTU screening. Algae dominated the plankton communities in both rivers. In the Chao River, 61 orders, 113 families, 223 genera, and 422 species were identified. The dominant genus was *Thalassiosira* (Bacillariophyta, 14.62%), followed by *Cyclotella* (Bacillariophyta, 7.76%) and *Mychonastes* (Chlorophyta, 6.84%) (Figure 3a). The Tuhai River exhibited slightly higher algal diversity, with 61 orders, 117 families, 222 genera, and 432 species. Dominant taxa included *Mychonastes* (Chlorophyta, 17.56%) and *Cyclotella* (Bacillariophyta, 6.7%), while other groups had relatively low abundances (Figure 3b). A total of 42 fish species were identified in the Chao River, with 11 orders, 18 families, and 34 genera. The genus *Carassius* dominated, representing 55.3% of total fish abundance, followed by *Cyprinus* (22.1%) and *Pseudorasbora* (11.9%) (Figure 3c). In the Tuhai River, 29 fish species were detected across 9 orders, 16 families, and 26 genera. *Planiliza* was the dominant genus (63.3%), followed by *Cyprinus* (9.34%) and *Oreochromis* (5.31%) (Figure 3d).

To further quantify the differences in community diversity between the two rivers, we calculated five alpha diversity indices for both the eukaryotic plankton and fish communities: the Shannon index, the Simpson index, Pielou’s evenness, Chao1, and observed species richness (Figure 4). Because these indices are derived from relative sequence read data, they are interpreted here as community-level proxies of compositional structure rather than precise quantitative measures of absolute abundance or biomass. No significant differences in α-diversity were observed for plankton communities (Figure 4a). The Shannon index (*p* = 0.27), the Simpson index (*p* = 0.70), Pielou’s evenness (*p* = 0.26), Chao1 (*p* = 0.34), and observed species richness (*p* = 0.13) did not differ significantly between the two rivers. For fish communities, diversity indices describing dominance structure did not differ significantly between rivers: the Shannon (*p* = 0.63) and Simpson indices (*p* = 0.14) were similar in the Chao and Tuhai Rivers (Figure 4b). In contrast, Pielou’s evenness (*p* = 0.007), Chao1 richness estimate (*p* = 0.00016), and observed richness (*p* = 0.00022) were all significantly higher in the Chao River than in the Tuhai River. These results suggest that although overall diversity levels were similar, the fish community in the Chao River exhibited greater evenness in species distribution and higher taxonomic richness. In contrast, planktonic communities showed greater consistency in both diversity and abundance between the two rivers.

### 3.3. β-Diversity and Community Assembly Processes

The beta diversity and community assembly processes of eukaryotic plankton and fish also showed notable differences between the two rivers. PCoA based on the Bray–Curtis distances revealed clear separation in the community structures of both eukaryotic plankton and fish between the Chao River and the Tuhai River (Figure 5). Spatial variation was more pronounced in plankton communities, with PCoA1 and PCoA2 explaining 15.64% and 11.82% of the variance, respectively (R^2^ = 0.40, *p* = 0.001) (Figure 5a). For fish communities, the first two PCoA axes accounted for 17.67% and 13.24% of the variation, respectively. The PERMANOVA results confirmed significant differences in community composition between the two rivers (R^2^ = 0.17, *p* = 0.001) (Figure 5b). These findings suggest that plankton communities display a stronger river-specific differentiation pattern in β-diversity compared to fish communities.

Additionally, both trophic-level communities displayed a clear distance–decay relationship (Figure 6), with community similarity decreasing significantly with increasing geographic distance. In contrast to fish communities, plankton communities showed steeper slopes: −0.00304 (R^2^ = 0.25, *p* < 0.001) in the Chao River and −0.00338 (R^2^ = 0.39, *p* < 0.001) in the Tuhai River (Figure 6a). For fish communities, the regression slope was −0.00265 (R^2^ = 0.12, *p* < 0.001) in the Chao River and −0.00232 (R^2^ = 0.11, *p* < 0.001) in the Tuhai River (Figure 6b). These results indicate that plankton communities exhibited significantly higher spatial turnover rates than fish communities across all trophic levels. While the distance–decay patterns of fish communities were similar between the two rivers, plankton communities showed stronger spatial turnover in the Tuhai River.

The NCM further revealed distinct assembly processes between the plankton and fish communities (Figure 7). The plankton communities exhibited a stronger fit to the neutral model, with R^2^ = 0.866 in the Chao River and R^2^ = 0.786 in the Tuhai River, suggesting that stochastic processes, such as ecological drift and dispersal limitation, played a dominant role in their assembly (Figure 7a,b). In contrast, the neutral model explained lower proportions of variance in fish communities: R^2^ = 0.322 with Nm = 332 in the Chao River, and R^2^ = 0.349 with Nm = 206 in the Tuhai River (Figure 7c,d). These results indicate that deterministic processes, including environmental filtering and biological interactions, likely exert a stronger influence on fish community assembly.

### 3.4. Environmental Responses Vary Across Different Trophic Levels

CCAs and Mantel tests were performed to identify key environmental factors influencing the structures of the plankton and fish communities in the two rivers (Figure 8). For fish communities, CCA revealed a clear separation of samples along environmental gradients, with the first two axes explaining 23.85% and 15.74% of the total variance, respectively. Six water quality parameters, namely EC, WT, pH, COD, BOD_5_, and DO, showed significant correlations with fish community composition (*p* < 0.05), indicating that these physicochemical factors play a dominant role in shaping the fish community structure. For plankton communities, CCA indicated stronger and more comprehensive environmental responses. The first two axes explained 21.69% and 14.23% of the variance, respectively. All nine measured water quality parameters, namely, EC, WT, TN, TP, pH, NH_4_-N, COD, BOD_5_, and DO, were significantly correlated with plankton structure (*p* < 0.05), demonstrating broader environmental sensitivity and a pronounced influence of nutrient gradients on plankton assembly.

Mantel tests further corroborated the above findings. Significant positive correlations were observed between community dissimilarity and environmental distance for both the fish and plankton communities in the two river systems (Figure 9). Among them, the plankton communities exhibited stronger correlations, as indicated by their higher Mantel r values, suggesting that water quality factors exerted a more substantial influence on plankton assembly. This finding aligns with plankton’s heightened sensitivity to environmental variation.

## 4. Discussion

Land use changes influence aquatic communities through both direct and indirect pathways. Direct effects primarily involve habitat loss or alteration due to land conversion, which subsequently modifies community structure [25]. Indirect effects occur when land use practices increase the input of pollutants into river ecosystems, leading to species migration or mortality due to the adverse effects of these contaminants [26]. In this study, differing land use types in the riparian buffer zones of two adjacent, similarly scaled rivers might result in the creation of distinct aquatic environments. The Tuhai River’s buffer zone was dominated by agricultural land, whereas the Chao River’s buffer zone was characterized by urban land use, which might contribute to the markedly different water quality conditions of the two systems. Using eDNA metabarcoding, this study examined biodiversity patterns, community assembly mechanisms, and environmental responses across multiple biological groups in riverine ecosystems. The results provide multi-trophic evidence supporting the “small-scale environmental selection hypothesis” [27,28].

### 4.1. Minimal Inter-River Variation in α-Diversity Despite Contrasting Environmental Conditions

Despite marked differences in environmental conditions such as land use patterns and water quality parameters between the Chao River and the Tuhai River, α-diversity exhibits only limited and metrologically characterized responses across different biological groups. This finding may be attributed to the inherent properties of α-diversity metrics. As a fundamental measure of biodiversity, the α-diversity reflects local species richness and evenness. However, it has limited capacity to capture niche differentiation among taxa or to reflect underlying community assembly processes, making it relatively insensitive to variations in the local environment [29,30]. Previous studies have shown that α-diversity may not accurately represent community responses to environmental gradients at broader watershed scales. For instance, Li et al. [31] compared multiple tributaries across a river system and observed no significant differences in α-diversity among them, suggesting that reliance on α-diversity alone may mask species-level turnover along environmental gradients. In river systems subject to pronounced environmental change, Li et al. [32] reported significant shifts in the α-diversity of both fish and algal communities, indicating that sensitivity may depend on the magnitude of environmental heterogeneity. Overall, this study suggests that for small-scale comparisons of adjacent rivers, α-diversity is not a sensitive indicator of differences in eukaryotic plankton and fish community structure. The α-diversity index serves as a benchmark descriptor for the richness and evenness of plankton and fish populations, providing a traditional framework for comparing community structures across trophic levels. Thus, it is still necessary to combine β-diversity to reveal differences in river biodiversity patterns.

### 4.2. Phytoplankton Communities Show Stronger River-Specific Differentiation and Spatial Turnover Under Contrasting Environmental Conditions

Both PCoA and distance–decay relationships revealed stronger β-diversity differentiation and higher spatial turnover in plankton communities between the Chao River and the Tuhai River compared to fish communities. This pattern may be linked to differences in niche characteristics and environmental response rates across trophic levels. Plankton, characterized by short life cycles and rapid population turnover, can respond quickly to fluctuations in water quality and nutrient availability [14,33]. From a metacommunity perspective, the rapid and intense response of plankton to local nutrient conditions exemplifies a classic process of “species sorting”: spatial variations in nitrogen, phosphorus, and other resources act as environmental filters, screening and shaping distinct planktonic assemblages at different locations [34]. As a result, they are more likely to undergo distinct community succession even between adjacent river systems. In contrast, fish, which have longer lifespans and greater dispersal capacity, tend to maintain more stable community structures and exhibit lower sensitivity to variation in the local environmental [35]. Frequent movement along river networks can produce large-scale effects, homogenizing fish communities across different locations. Notably, plankton communities in the Tuhai River showed even stronger β-diversity and spatial turnover than those in the Chao River. This may be explained by the higher nutrient levels and intensified anthropogenic pressures in the Tuhai River. Nutrients and seasonal variations are known to be key drivers of plankton assembly, while physicochemical parameters such as temperature directly regulate metabolic rates and community dynamics [36]. These combined stressors may have reduced the influence of stochastic processes in the Tuhai River plankton community, thereby amplifying its spatial differentiation and further strengthened nutrient-driven species selection at the small spatial scale.

### 4.3. In Small-Scale Watersheds, Stochastic and Deterministic Processes, Respectively, Dominate Plankton and Fish Community Assembly

Both deterministic and stochastic processes collectively shape aquatic community assembly, though their relative contributions vary across communities [37]. The NCM-based results of this study suggest that stochastic processes dominate plankton assembly. This aligns with previous findings on planktonic organisms. Lu et al. [38] reported that stochasticity drives algal community structure in the rivers and lakes of the Qinghai–Tibet Plateau, and Wang et al. [11] observed a strong influence of stochasticity in eukaryotic plankton across multiple river systems, including the Yongding, Beiyun, Ting, Jinsha, and Lancang Rivers. These findings highlight fluctuations in environmental conditions such as temperature and hydrological dynamics, creating a more unpredictable environment for plankton and allowing random processes like random diffusion to dominate community formation. In contrast, the lower fit of the NCM for fish communities indicates the greater role of deterministic processes in their assembly. Li et al. [39] highlighted that human activities can intensify deterministic filtering in fish communities, consistent with their strong mobility and specific niche requirements. Lu et al. [38] further argued that vertebrates, which are capable of undertaking long-distance migration in search of suitable habitats and food sources, are more influenced by deterministic factors such as environmental filtering and species interactions. It is worth noting that at broader spatial scales, such as estuarine regions, fish assembly may exhibit higher stochasticity, especially among non-native species, due to dynamic hydrological conditions and frequent disturbances [40]. Overall, given the geographic proximity of the Chao and Tuhai Rivers, fish can disperse freely under similar regional conditions; however, the local community structure appears to be strongly shaped by deterministic processes linked to niche partitioning. Despite the connectivity between these two rivers, local environmental factors, such as habitat suitability and species interactions, play a more significant role in structuring the fish communities, reinforcing the importance of deterministic processes in shaping higher trophic levels. Taken together, these differences between biological groups refine the “small-scale environmental selection hypothesis” by showing that its strength and manifestation depend on organismal traits and trophic position.

### 4.4. Different Biological Communities Exhibit Varying Sensitivities and Response Ranges to Aquatic Environmental Conditions

Although neutral processes significantly contribute to community assembly, particularly for plankton, our findings demonstrate that the influence of local environmental factors on taxonomic composition varies along environmental gradients. CCA and the Mantel test revealed that plankton communities are strongly associated with nutrient factors such as TP and TN. Nitrogen and phosphorus are essential drivers of algal growth, and fluctuations in their concentrations can markedly affect algal growth rates, biomass, and population structure, thereby indirectly affecting eukaryotic plankton abundance [41]. In contrast, fish communities are more closely linked to physicochemical variables, including EC, pH, and COD, consistent with their known sensitivity to changes in temperature and dissolved oxygen [42,43]. The Chao River and the Tuhai River, though geographically adjacent and both subject to environmental pollution, exhibit distinct water quality and nutrient profiles. These differences establish a clear environmental gradient, in turn eliciting trophic-level-specific ecological responses. Previous studies have shown that water pollution-related factors are key determinants of aquatic community structure at both the river and watershed scales [27,28]. Our study further refines and supports the “small-scale environmental selection hypothesis,” highlighting that the type and intensity of environmental filtering vary across trophic levels, even within localized ecosystems.

## 5. Conclusions

This study utilized eDNA metabarcoding to compare aquatic communities in two geographically adjacent rivers with distinct riparian land use and aquatic environmental characteristics. By jointly analyzing α-diversity, β-diversity, distance–decay relationships, and community assembly mechanisms across biological groups, we show that river-specific differences in biodiversity are primarily expressed through spatial turnover and assembly processes rather than uniform variations in local diversity. For eukaryotic plankton, α-diversity indices did not differ between the Chao and Tuhai Rivers, whereas for fish, richness- and evenness-based indices were higher in the Chao River. In contrast, both plankton and fish exhibited clear river-specific differentiation in β-diversity, and plankton communities displayed stronger river-based differentiation and steeper distance–decay relationships than fish. Our results reveal pronounced biological groups differences in community assembly. Neutral Community Model fits indicate that stochastic processes such as ecological drift dominate plankton assembly; however, the strong correlations between plankton community composition and nutrient variables (TP, TN, NH_4_–N) point to nutrient-driven species sorting being superimposed on this stochastic background. For fish communities, the low NCM fit and stronger associations with physicochemical parameters (e.g., EC, pH, COD) indicate that deterministic processes such as environmental filtering and niche partitioning play a more substantial role in structuring local assemblages under conditions of high dispersal connectivity between the two rivers. The differences between these biological groups demonstrate that different components of biota experience distinct degrees and modes of “small-scale environmental selection” in river ecosystems. Taken together, this multi-trophic analysis refines and extends the “small-scale environmental selection” hypothesis originally proposed for zooplankton in polluted rivers. Our findings confirm that, at fine spatial scales, pollution-induced environmental gradients can override dispersal and act as key drivers of community structure, as well as showing that the expression of small-scale environmental selection is trait- and biological group-dependent. Finally, this study underscores the value of eDNA metabarcoding for multi-trophic community characterization and highlights the importance of including biological group analysis within aquatic ecosystem assessment and management within the context of ongoing land use and water quality change.

## Figures and Tables

**Figure 1 biology-14-01796-f001:**
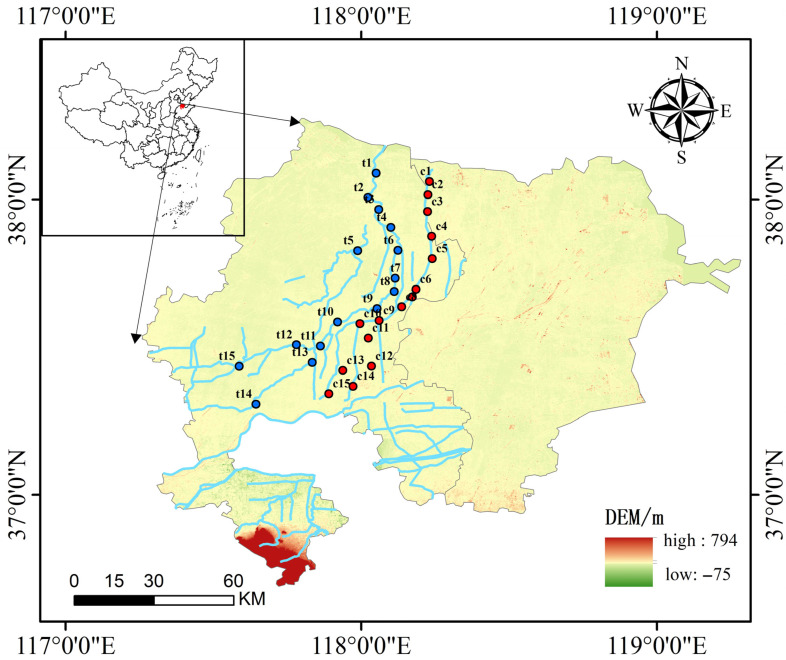
The location of the study area and distribution of sampling sites. Red indicates sampling points located on the Chao River; blue indicates sampling points located on the Tuhai River. Elevation data source: National Earth System Science Data Center (China).

**Figure 2 biology-14-01796-f002:**
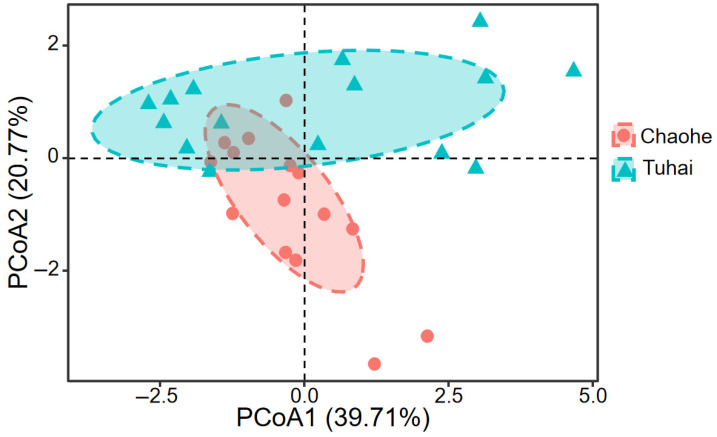
PCoA results for water quality differences between the adjacent rivers.

**Figure 3 biology-14-01796-f003:**
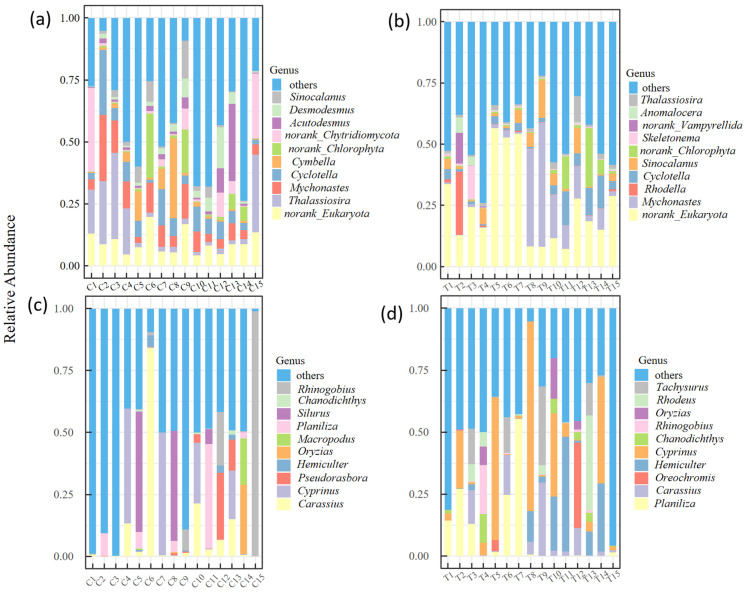
The relative sequence abundances of the top 10 dominant species at the genus level. (**a**,**b**) The relative abundances of plankton species in the Chao River and the Tuhai River. (**c**,**d**) The relative abundances of fish species in the Chao River and the Tuhai River.

**Figure 4 biology-14-01796-f004:**
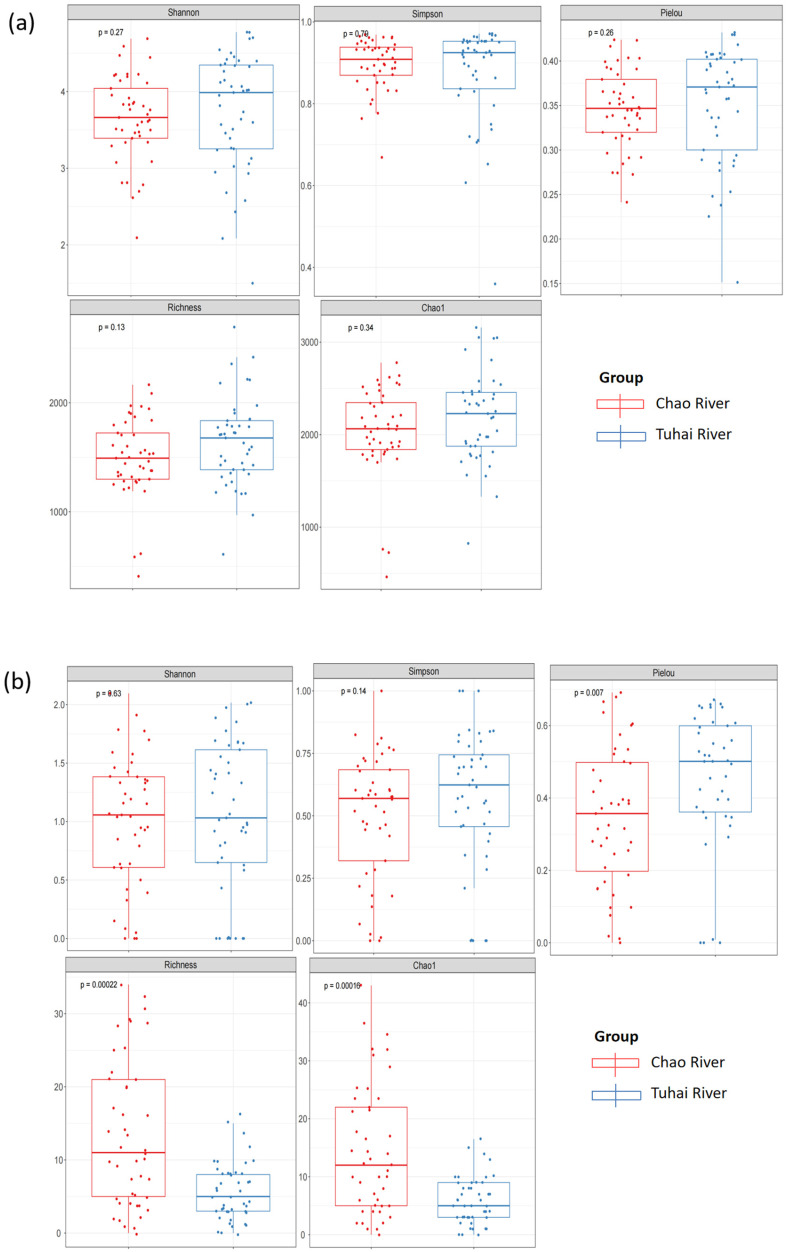
Alpha diversity indices of plankton and fish communities in the Chao River and the Tuhai River. (**a**) Plankton alpha diversity indices. (**b**) Fish alpha diversity indices.

**Figure 5 biology-14-01796-f005:**
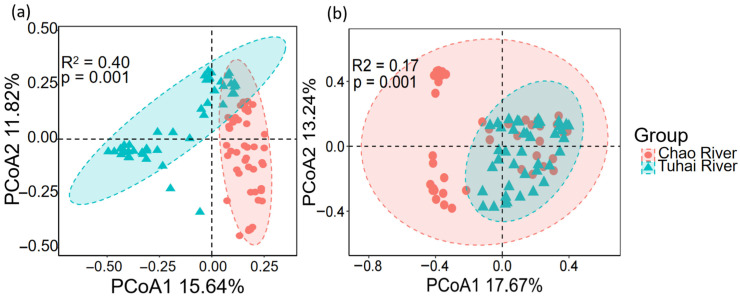
PCoA of communities at different trophic levels in the Chao River and the Tuhai River: (**a**) the plankton community and (**b**) the fish community.

**Figure 6 biology-14-01796-f006:**
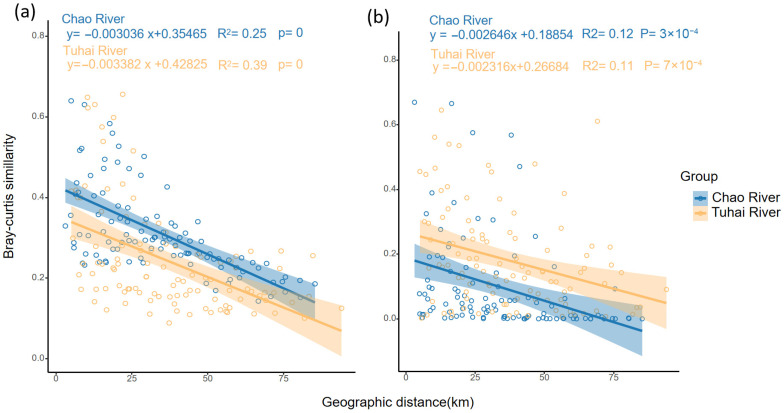
The distance–decay relationships for plankton and fish communities in the Chao River and the Tuhai River. (**a**) The distance–decay patterns of plankton communities. (**b**) The distance–decay patterns of fish communities. Pearson correlation coefficients (R) and *p*-values are indicated. The dots represent pairwise community similarity between sampling sites, and the shaded areas represent 95% confidence intervals around the regression lines.

**Figure 7 biology-14-01796-f007:**
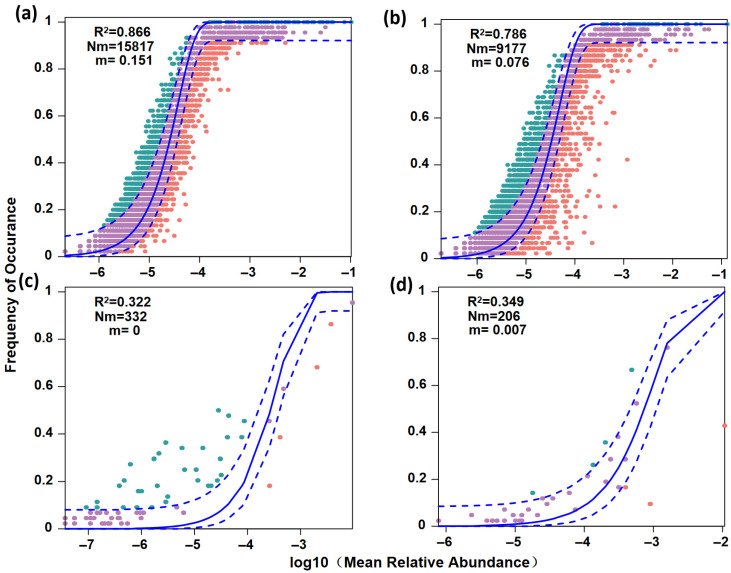
The Neutral Community Model fitting for plankton and fish communities. (**a**) The plankton community in the Chao River. (**b**) The plankton community in the Tuhai River. (**c**) The fish community in the Chao River. (**d**) The fish community in the Tuhai River. Factors occurring more or less frequently than predicted using the neutral model are shown in green and pink, respectively.Purple dots indicate taxa that fit the neutral model prediction. Dashed lines represent 95% confidence intervals around the model prediction (solid line). R^2^ values indicate the model’s fit and m indicates the estimated migration rate.

**Figure 8 biology-14-01796-f008:**
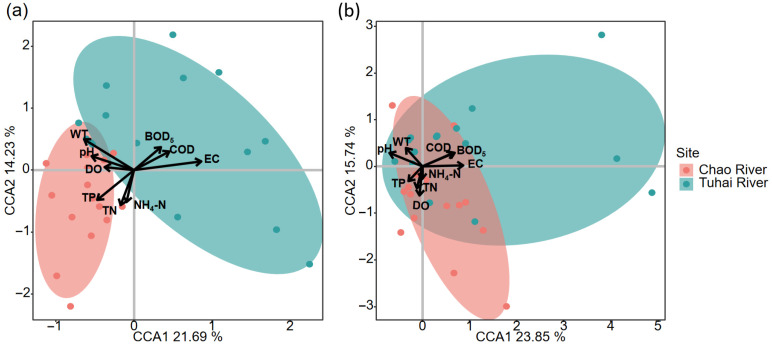
CCA of species–environment relationships (**a**) plankton community and (**b**) fish community. Arrows indicate environmental factors; points represent sampling sites.

**Figure 9 biology-14-01796-f009:**
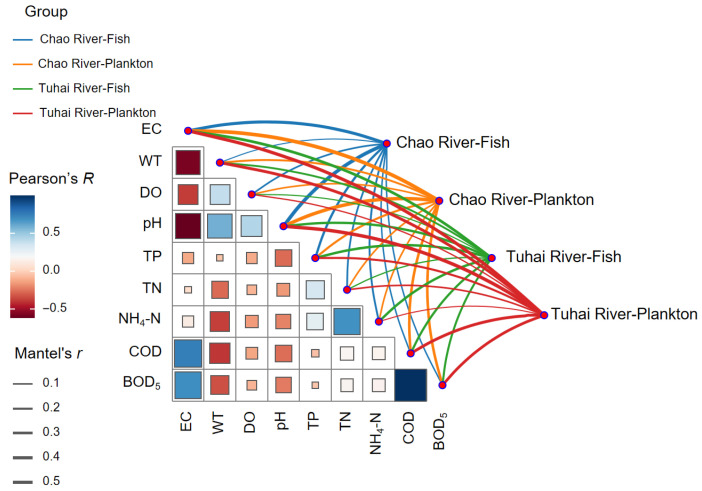
The Mantel test results of correlations between community composition and spatial and environmental factors. Edge width corresponds to Mantel r values; edge color indicates statistical significance. The color gradient in the background matrix represents pairwise Pearson correlations among spatial and environmental factors.

## Data Availability

The original contributions presented in this study are included in the article/Appendix A. Further inquiries can be directed to the corresponding author.

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
