# Peer review of "Environmental DNA Metabarcoding Reveals Divergent Patterns of Biodiversity, Community Assembly, and Environmental Sensitivity Across Taxa in Adjacent Rivers"

_biology, 2025, doi:10.3390/biology14121796_

Round 1

Reviewer 1 Report

Comments and Suggestions for Authors

This manuscript leverages a powerful, modern tool (eDNA metabarcoding) to address a fundamental question in river ecology: how different trophic levels (plankton and fish) differentially respond to environmental drivers and are assembled by distinct processes. The study's central finding—that plankton assembly is predominantly stochastic and nutrient-sensitive, while fish assembly is deterministic and nutrient-insensitive—is provocative and conceptually compelling. However, the study is critically undermined by a fatal methodological flaw: the sampling design relies on a single point in time ("April 2024"). It is scientifically untenable to draw robust conclusions about fundamental ecological "mechanisms" or "processes" from a single snapshot, especially given the highly dynamic nature of plankton (daily turnover) and the seasonal behavior of fish (e.g., migrations). Furthermore, the manuscript fails to provide a mechanistic explanation for the apparent paradox of how fish can be insensitive to the very nutrient gradients that are reported to strongly structure their own food base.

The study's most important and novel contribution is the empirical, eDNA-based demonstration of a sharp dichotomy in assembly patterns and environmental drivers between eukaryotic plankton and fish in adjacent, environmentally distinct river systems.

The central idea is excellent and the preliminary data are intriguing. However, the conclusion that these observed differences are due to fundamental "assembly mechanisms" is an unjustifiable overextrapolation from a single-time-point sampling design. The patterns observed in April could be an ephemeral artifact of spring conditions (e.g., a stochastic plankton bloom and a deterministic fish spawning run). To be considered for a high-impact journal, the authors must demonstrate the temporal stability of these patterns. This would require, at minimum, resampling across multiple seasons (e.g., wet vs. dry seasons, or all four seasons) to validate that the observed assembly mechanisms are a fundamental feature of these rivers and not a transient event.

Detailed section-by-section analysis:

  1. Introduction

Strengths:

Effectively establishes the broad context of river ecosystem degradation by land use. Clearly introduces the competing theoretical frameworks (niche vs. neutral theory) that are central to the paper. Identifies a clear knowledge gap: the lack of comparative studies on assembly mechanisms and environmental responses across multiple trophic levels.

Weaknesses and Suggestions for Improvement. Functional Ambiguity: As in the abstract, the introduction discusses "eukaryotic plankton" by grouping phytoplankton (producers) and zooplankton (consumers). This is problematic as they occupy different trophic levels. Action: The introduction must operationally define what "eukaryotic plankton" means in this study (e.g., "we focus on eukaryotic plankton, which in these rivers is comprised predominantly of primary producers...") or justify why analyzing this functional mixture as a single group is valid.

2. Methodology / Materials and Methods

About the question: Is the research design appropriate?

No, the research design is fundamentally inappropriate for achieving its primary stated goals.

The study aims to make strong claims about "community assembly mechanisms" and "differential environmental selection processes". However, the design is a single-point-in-time snapshot ("Sampling was conducted in April 2024" ), which cannot be used to validly infer long-term processes.

There's a fatal red flaw: "Snapshot" vs. "Process" The entire study's conclusions about how communities are assembled (stochastic vs. deterministic) are built on a single sampling event. This is an invalid design for the following reasons:

Plankton dynamics: The "eukaryotic plankton" community, particularly algae, has an extremely rapid turnover rate (days to weeks). The community structure observed in April is a transient snapshot that may be completely different from the community in July or October. It is not a stable "process."

Fish seasonality: fish communities are highly seasonal. Their presence, abundance, and distribution are dictated by temperature, spawning migrations, and feeding patterns. The "deterministic" (non-random) pattern observed in the fish community in April  could simply be an artifact of a spring spawning run, not a year-round assembly rule.

A design claiming to test processes or mechanisms must include a temporal component, such as sampling across multiple, distinct seasons (e.g., spring, summer, fall, winter, or at minimum, a dry vs. wet season) to demonstrate that the observed patterns are stable and not just an artifact of one specific time of year.

Beyond the later, the design has other weaknesses that compromise its conclusions:

Ambiguous trophic grouping: the study claims to compare "trophic levels", but it creates a functionally ambiguous group. "Eukaryotic plankton" explicitly includes both primary producers (phytoplankton) and primary consumers (zooplankton). By lumping these distinct trophic levels into one group using a single 18S marker, the design is not making a clean comparison between "producers" and "top consumers" (fish).

Insufficient sample volume for fish: the design uses a 1-liter water sample for eDNA analysis. While this volume may be adequate for capturing the dense plankton community, it is very small for assessing fish. Fish eDNA is often much scarcer and more patchily distributed. This small volume increases the risk of false negatives (failing to detect fish that are present), which would lead to an inaccurate measure of fish community structure and unreliable comparisons to the plankton community.

Inaccurate richness assessment: the design relies on 12S and 18S metabarcoding, which is effective for detecting presence but is notoriously unreliable for estimating true abundance or biomass. Despite this, the study calculates "Relative Abundance" (Fig. 3) and uses abundance-based metrics (Pielou's Evenness, Shannon, Simpson). This is a well-known limitation of eDNA, and the design does not account for the high potential of PCR bias and primer mismatch skewing these "abundance" numbers.

About the question: Are the methods adequately described?

No, the methods are not adequately described. While the manuscript provides commendable, specific detail on the wet-lab PCR protocols and field sampling, it critically fails to describe the bioinformatics and quality control pipeline. This section is a "black box" that makes the results completely non-reproducible, which is an immediate red flag for any high-impact journal.

The study is unpublishable in its current form due to these major gaps in the methods:

No mention of negative controls: this is the most severe flaw. For any eDNA study, the inclusion of field blanks, extraction blanks, and PCR negative controls (NTCs) is non-negotiable to detect and control for contamination. The methods do not mention them at all. While the results section vaguely states "Non-target sequences such as environmental background contaminants were excluded", the method for how this was done (e.g., subtracting reads from blanks) is completely absent.

Inadequate bioinformatics pipeline: This section is critically underdeveloped and lacks the necessary detail for reproduction.

Quality filtering: The only filtering step mentioned is discarding reads shorter than 200 bp. There is no mention of industry-standard Phred score (Q-score) filtering to remove low-quality base pairs or reads.

Chimera removal: The protocol omits any mention of chimera detection and removal (e.g., via DADA2, UCHIME, or VSEARCH). This is an essential, standard step, and its absence means the OTU and richness data are likely inflated with artifacts.

OTU clustering: The paper states sequences were "clustered into Operational Taxonomic Units (OTUs)"  but fails to name the algorithm (e.g., UPARSE, VSEARCH) or the similarity threshold (e.g., 97%, 99%). This single parameter fundamentally changes all downstream richness and diversity metrics.

Taxonomic assignment: The paper lists the databases (SILVA, MitoFish)  but not the assignment algorithm (e.g., BLASTn, RDP classifier) or the confidence/identity threshold (e.g., 97% identity, 90% confidence) used to assign a name to an OTU.

About the question: Are the results clearly presented?

No, the results are not clearly presented. While most of the section is logically structured, it contains a critical internal contradiction that fundamentally misrepresents the data. This is a severe weak that undermines the credibility of the authors' observations.

"Community composition and a-diversity," the manuscript directly contradicts itself when discussing the fish community data.The Claim: The text states, "For fish communities, neither the Shannon (p=0.63) nor the Simpson index (p=0.14) showed significant differences between the Chao and Tuhai Rivers". The Contradiction: In the very next sentence, the text states, "However, Pielou's evenness (p=0.007), Chao richness estimate (p=0.00016), and observed richness (p=0.00022) were all significantly higher in the Chao River". An author cannot claim "no significant differences" and then immediately report three highly significant p-values for 3 out of 5 metrics. This contradictory presentation makes it unclear if the authors have misunderstood their own results or are attempting to downplay findings that may not fit their narrative. This error is confirmed by visually inspecting Figure 4b, which clearly shows significant differences.

About the question: Are the conclusions supported by the results?

No, the conclusions are not fully supported by the results because they contain a major internal contradiction. While most of the conclusions are well-supported, a key claim about alpha-diversity is directly refuted by the data presented in the results section.

The conclusion states that "...alpha-diversity and species richness of eukaryotic plankton and fish communities showed no significant differences between the two rivers". This is factually incorrect. The Results section explicitly states: "For fish communities, neither the Shannon (p=0.63) nor the Simpson index (p=0.14) showed significant differences...". "However, Pielou's evenness (p=0.007), Chao richness estimate (p=0.00016), and observed richness (p=0.00022) were all significantly higher in the Chao River". The authors cannot claim "no significant differences" in the conclusion when their own results show that 3 out of 5 metrics were, in fact, highly significant. This major error is problematic because the other main conclusions of the paper are well-supported by the results:

Beta-Diversity: The conclusion that beta-diversity showed "clear river-specific differentiation" is supported. The PCoA plots and PERMANOVA tests showed clear, significant separation for both plankton (R2=0.40, p=0.001) and fish (R2=0.17, p=0.001). Assembly Mechanisms: The conclusion that stochastic processes dominated plankton assembly while deterministic processes structured fish communities is supported. The Neutral Community Model (NCM) showed a very high fit for plankton (R2 > 0.78) and a very low fit for fish (R2< 0.35). Environmental Sensitivity: The conclusion that plankton were more sensitive to nutrient gradients is supported. The CCA and Mantel tests showed plankton were significantly correlated with all nine environmental factors, including nutrients (TP, TN, NH4-N), while fish were not correlated with those nutrient factors.

Comments on the Quality of English Language

The English is functional and understandable but lacks the exceptional clarity, conciseness, and persuasive force expected in a top-tier journal. There are repetitive phrases and some slightly awkward constructions (e.g., "a dearth of research exists").

The manuscript would benefit significantly from professional copy-editing by a native English speaker with a background in ecology. Or use the MDPI english-editing services. 

Reviewer 2 Report

Comments and Suggestions for Authors

This manuscript presents an interesting and valuable contribution on trophic-level differences in community assembly and environmental responses, based on eDNA metabarcoding of two adjacent river systems. However, the manuscript requires substantial improvement in both scientific depth and presentation quality. Conceptually, several sections remain overly descriptive and repetitive, with limited discussion of the underlying ecological mechanisms and theoretical implications.

In addition, the manuscript would benefit from careful linguistic and structural refinement. The writing should be streamlined to minimize redundancy and enhance readability. A professional language revision would be recommended to improve clarity, coherence, and academic tone. Detailed comments are as follows:

  1. Abbreviations (e.g., EC, WT, DO, COD, BOD₅) appear without definition. Please spell them out at first mention in the abstract.
  2. The abstract is too long; please condense it to highlight key objectives, methods, and main findings more concisely.
  3. The introduction provides a solid background but remains too general and diffuse. It would benefit from a clearer focus and more concise structure. The authors should streamline the description on global land-use impacts and move more directly to the specific research problem. The knowledge gap—how community assembly and environmental responses differ across trophic levels—needs clearer definition and stronger justification for its scientific importance. The link between niche/neutral theory and the study’s objectives should be explicitly stated to show how theory supports the comparative approach.

In addition, the introduction omits description of the study area. A brief overview of the ecological characteristics, land-use context, and prior studies of the Tuhai and Chao Rivers would provide essential background and help readers understand the rationale for site selection. The final paragraph should also be revised to present the study objectives and to end with a concise statement of the study’s novelty and broader ecological significance.

  1. Figure1: Although the caption mentions the Chao and Tuhai Rivers, their main courses are not shown or labeled on the map. Adding and labeling these rivers would greatly improve geographic clarity and help readers interpret the sampling site distribution.
  2. The figure1 shows elevation, but the data source is not specified. Please indicate where the elevation data were obtained from.
  3. Line 131: The rationale for selecting a 2-km buffer radius is not explained.
  4. Line 136-142: Although the measurement methods for various water quality parameters are described in detail, no corresponding standard or authoritative references are cited. It is recommended that the authors include appropriate references to enhance the scientific rigor and reproducibility of the methods section.
  5. I would suggest improving the labels of Figures 4 and 6 to make them clearer and consistent with the labeling style used in other figures in the paper.
  6. Line 143: It is recommended to present this section in a table rather than long text to enhance clarity and readability.
  7. Line 187: It is recommended that all classical methods, analytical procedures, and indices described in the Materials and Methods section be supported by appropriate citations to their original or authoritative sources.
  8. Line 199: It is recommended to briefly mention the model’s assumptions and the basic principles of the classical ecological methods used to improve clarity and completeness.
  9. I recommend using “(a), (b), …” to label all figure panels for better clarity and consistency.
  10. Please check and ensure that all instances of ‘R²’ are written consistently and correctly throughout the paper.
  11. Line 369: The authors should clarify the purpose of including α-diversity analysis, given its low sensitivity at small scales. Explaining how it complements β-diversity or serves as a baseline for community comparison would strengthen the study’s rationale.
  12. The interpretation of β-diversity and spatial turnover is sound but overly descriptive. The authors should discuss the underlying ecological mechanisms - for instance, why phytoplankton respond more strongly to nutrient fluctuations and disturbance frequency, whereas fish communities show greater spatial homogenization driven by dispersal and environmental filtering. Incorporating theoretical frameworks such as species sorting and mass effects would improve the conceptual depth and rigor of this section.
  13. Line 430: This section remains descriptive. The authors are encouraged to briefly elaborate on the ecological mechanisms behind trophic-level differences—linking plankton responses to nutrient-driven species sorting and fish responses to physicochemical filtering-and to clarify how your findings refine or extend the “small-scale environmental selection hypothesis”.
  14. Line 450: The authors should streamline the Conclusion section by reducing repetition and emphasizing its broader ecological implications. In particular, I would suggest the authors to highlight how the study advances understanding of trophic-level differences in community assembly and refines the “small-scale environmental selection” concept.

Round 2

Reviewer 1 Report

Comments and Suggestions for Authors

Well done!. Now is a ready to publish. congrats!

Comments on the Quality of English Language

The English is functional and understandable but lacks the exceptional clarity, conciseness, and persuasive force expected in a top-tier journal. There are repetitive phrases and some slightly awkward constructions (e.g., "a dearth of research exists").

The manuscript would benefit significantly from professional copy-editing by a native English speaker with a background in ecology. Or use the MDPI english-editing services. 

Reviewer 2 Report

Comments and Suggestions for Authors

I am satisfied with the revision.